

# Sorption to soil, biochar and compost: is prediction to multicomponent mixtures possible based on single sorbent measurements?

Melanie Kah[1,2], Gabriel Sigmund[1,3], Pedro Luis Manga Chavez[1], Lucie Bielská[4] and Thilo Hofmann[1]

[1] Department of Environmental Geosciences, University of Vienna, Vienna, Austria
[2] Commonwealth Scientific and Industrial Research Organisation, Glen Osmond, SA, Australia
[3] Current affiliation: Chair of Analytical Chemistry and Water Chemistry, Technical University of Munich, Munich, Germany
[4] RECETOX, Faculty of Science, Masaryk University, Brno, Czech Republic

Corresponding authors
Melanie Kah,
melanie.kah@univie.ac.at
Thilo Hofmann,
thilo.hofmann@univie.ac.at

## ABSTRACT

Amendment with biochar and/or compost has been proposed as a strategy to remediate soil contaminated with low levels of polycyclic aromatic hydrocarbons. The strong sorption potential of biochar can help sequestering contaminants while the compost may promote their degradation. An improved understanding of how sorption evolves upon soil amendment is an essential step towards the implementation of the approach. The present study reports on the sorption of pyrene to two soils, four biochars and one compost. Detailed isotherm analyzes across a wide range of concentration confirmed that soil amendments can significantly increase the sorption of pyrene. Comparisons of data obtained by a classical batch and a passive sampling method suggest that dissolved organic matter did not play a significant role on the sorption of pyrene. The addition of 10% compost to soil led to a moderate increase in sorption (<2-fold), which could be well predicted based on measurements of sorption to the individual components. Hence, our result suggest that the sorption of pyrene to soil and compost can be relatively well approximated by an additive process. The addition of 5% biochar to soil (with or without compost) led to a major increase in the sorption of pyrene (2.5–4.7-fold), which was, however, much smaller than that suggested based on the sorption measured on the three individual components. Results suggest that the strong sorption to the biochar was attenuated by up to 80% in the presence of soil and compost, much likely due to surface and pore blockage. Results were very similar in the two soils considered, and collectively suggest that combined amendments with compost and biochar may be a useful approach to remediate soils with low levels of contamination. Further studies carried out in more realistic settings and over longer periods of time are the next step to evaluate the long term viability of remediation approaches based on biochar amendments.

Subjects Soil Science, Environmental Contamination and Remediation
Keywords PAH, Pyrene, Adsorption, Soil remediation, Biochar, Compost, Amendment, Isotherm, DOC, Dissolved organic matter

# INTRODUCTION

Biochar is a carbon rich material produced by the pyrolysis of a variety of feedstock at temperatures below 700 °C (*EBC, 2012*; *IBI, 2015*). Biochar was originally produced as a soil amendment with objectives including carbon sequestration, soil fertility improvement and by-product/waste recycling (*Ahmad et al., 2014*). Many studies have shown that biochar can exhibit a high sorption affinity for a range of organic contaminants including polycyclic aromatic hydrocarbons (*Cornelissen et al., 2005*; *Hale et al., 2011*; *Kah et al., 2016*), polychlorinated biphenyls (*Chai et al., 2012*; *Wang et al., 2013*; *Denyes, Rutter & Zeeb, 2013*) and ionisable compounds (*Teixidó et al., 2011*; *Xiao & Pignatello, 2015*; *Sigmund et al., 2016*). Sorption is a key process to limit the bioavailability and transport of organic contaminants, and the incorporation of biochar into contaminated soil and sediment has thus been suggested as a remediation strategy to limit exposure and off-site transport of contaminants from contaminated sites (*Ahmad et al., 2014*).

Ideally, a remediation strategy does not only aim at sequestering and reducing the uptake of organic contaminants, but also considers their possible degradation into harmless products. The presence of biochar can reduce the degradation kinetics of organic contaminants in soils (*Marchal et al., 2013*), and the simultaneous addition of compost has been considered to counteract the undesired impact of biochar on degradation, while providing an additional source of organic matter. In particular, two recent studies have suggested that combining biochar and compost amendments to soils contaminated with polycyclic aromatic hydrocarbons may be successful to significantly reduce toxicity to soil organisms e.g. *Caenorhabditis elegans* (*Bielská et al., 2017*), while avoiding the full inhibition of the degradation process (*Sigmund et al., 2018*). Both studies suggested that changes in toxicity and degradation occurred through changes in the contaminant sorption, but details regarding the process were limited to data obtained at relatively high concentrations and/or in the presence of multiple contaminants. More detailed investigations into the sorption process are thus needed to support the future development of remediation approaches combining biochar and compost amendments.

With the aim to gain more insights into the changes in sorption provoked by the addition of combined amendments, the present study considers detailed investigations into the sorption characteristics of a series of soil, biochar and compost mixtures. Pyrene was selected as a model organic contaminant representative of polycyclic aromatic hydrocarbons. The study design directly builds on the results of *Bielská et al. (2017)* and *Sigmund et al. (2018)*, and focuses on aspects identified as requiring further investigations.

Sorption was first measured on four different types of biochars produced from miscanthus and soft wood, either at 550 or 700 °C. The biochar with the highest sorption potential was further considered to systematically investigate sorption in two soils, one compost and one biochar, either alone or in two and three-phase mixtures. Soil, compost and biochar are expected to interact upon mixing, and the sorption properties of their mixture is likely to differ from the sum of their individual sorption properties. In particular, the strong sorption of biochar is expected to be reduced through surface

fouling by soil and/or compost. One of the main objective of this study was to determine the extent to which sorption in mixtures deviates from a simple additive model. Another objective was to consider sorption across a wide range of concentration, and to distinguish the contributions of partitioning (into labile organic matter) and adsorption (at the surface of carbonised surfaces of biochar) by carrying out an in-depth isotherm analysis.

Our study was also designed to consider the possible impact of dissolved organic matter on sorption. Compost is typically very rich in organic matter (>20%, *Brinton, 2000*), and may produce dissolved organic matter that can facilitate the transport and uptake of sorbed contaminants. For instance, the results of *Bielská et al. (2017)* indicated a possible contribution of particle-bound pathways to toxicity. Sorption is typically measured using the batch method, which includes a phase separation step often based on centrifugation. Dissolved organic matter may not be fully separated from the aqueous phase by centrifugation. Conversely, passive sampling methods such as that based on polyoxymethylene (POM), only consider the freely dissolved fraction of pyrene and eliminate the possible contribution of dissolved organic matter on sorption. In the present study, the sorption of pyrene was measured with the batch method as well as by passive sampling. Comparing sorption coefficients derived by the two methods indicates whether dissolved organic matter plays a significant role on sorption.

Overall, our study aimed at gaining a better understanding of the sorption of pyrene in mixtures of soil, biochar and compost, across a wide range of contaminant concentrations, and accounting for the possible effect of dissolved organic matter. These aspects are key to the further development and implementation of remediation strategies based on combined amendments but they have not been systematically considered up to now.

## MATERIALS AND METHODS

### Sorbents

Biochars produced from soft wood and miscanthus grass straw at pyrolysis temperatures of 550 and 700 °C were purchased from the UK Biochar Research Centre (SWP550, SWP700, MSP550 and MSP700, respectively). A compost containing 11.4% organic carbon was kindly provided by fk Agrar Umweltservice (Pixendorf, Austria). Two top soils were sampled from agricultural fields in Austria: a sandy loam from Eschenau (Lower Austria) and a clay loam from Kaindorf (Styria). After sampling, the two soils were air dried, sieved at <2 mm and stored in the dark. Soil samples were amended with compost and/or biochar (all air dried, see the details in the sorption section), well mixed and conditioned for about three days at 20 °C. The mixtures were then dried overnight at 80 °C, crushed with a pestle and mortar, and sieved at <250 μm to ensure homogenisation before the sorption experiments. All materials were thoroughly characterized in previous studies, including elemental analysis, organic carbon, pH, specific surface area and pore volume (*Sigmund et al., 2017b*; *Bielská et al., 2017*). All properties are presented in Tables S1–S3 of the supporting information. Scanning electron microscope images were obtained for the four biochars with an Inspect™ S50 scanning electron microscope

operating under high vacuum, 10 kV acceleration voltage equipped with a secondary electron Everhart–Thornley detector (Figs. S1 and S2) following sputtering of the samples with carbon.

## Chemicals and analysis

Hexane and methanol were of residue analysis grade (Labscan, Dublin, Ireland and Acros Organics, Geel, Belgium). Pyrene (99.5%) and pyrene-d10 (99.5%) were purchased from Dr. Ehrenstorfer GmbH (Augsburg, Germany). Pyrene was analyzed by gas chromatography-mass spectrometry (GC-MS): Agilent 7890A coupled to Agilent 5975C; HP-5MS fused silica column: 60 m $\times$ 250 $\mu$m $\times$ 0.25 $\mu$m, J&W Scientific; pulsed splitless mode; oven temperature of 55 °C for 1 min, then 10 °C/min up to 300 °C. Quantification was based on deuterated internal standards added to the samples before extraction. The limit of detection was 0.0054 $\mu$g/L for the batch sorption experiment and 0.0002 $\mu$g/L for the passive sampling method (the concentration refers to that remaining in the aqueous phase of the sorbent suspension after sorption).

## Sorption measurements

Sorption of pyrene was first measured on the four biochars (SWP550, SWP700, MSP550 and MSP700) and the compost. The biochar exhibiting the highest sorption potential was further used in the sorbent mixtures: compost + biochar (2:1) and for each soil: soil + 10% compost, soil + 5% biochar, and soil + 10% compost + 5% biochar (all in dry weight based). The rates were selected to be consistent with previous studies considering combined amendments (*Bielská et al., 2017*; *Sigmund et al., 2018*). Sorption was measured using two techniques: a classical batch and a passive sampling method using POM. All sorption experiments were conducted at 20 °C and using aqueous background solutions containing 0.01 M $CaCl_2$ and 0.385 mM $NaN_3$ to inhibit biological activity. Pyrene stock solutions were prepared in methanol, and the amount spiked was kept below 0.1% in volume to minimize cosolvent effects on sorption. Controls and blanks were included in all experiments and the sorbed concentrations were calculated by mass balance.

For the batch method, sorbent suspensions were prepared in 50 mL glass centrifuge tubes (10 mg of biochar, 20 mg of compost, 200 mg of soil or soil mixture in 40 mL of background solution) and equilibrated for 48 h by horizontal shaking at 125 rpm. Pyrene was then spiked (the initial concentration ranged 1–50 $\mu$g/L), before samples were return to shaking for 48 h, which was previously demonstrated to be sufficient to reach sorption pseudo-equilibrium (data not shown). Samples were then centrifuged at 1000 $g$ for 40 min, 30 mL of supernatant were collected, and extracted three times with hexane after addition of the internal standard (Pyr-$d_{10}$). Extracts were combined, concentrated down to 1 mL under $N_2$ flow and analyzed by GC-MS (see the details above). Sorption was measured by batch at six different concentration levels for each sorbent and in triplicates.

The POM method allows sampling the truly dissolved portion of pyrene and studying the possible effects that dissolved organic matter may have on sorption measurements. The POM method also allows measuring sorption down to lower concentrations than
with the classical batch. The method was previously described and validated for pyrene (*Kah et al., 2011*, *2016*) and it is only briefly described here. Clean POM strips of about 100 mg were preconditioned in the background solution for 72 h, before being added to sorbent suspensions prepared as described for the batch method. The samples were then spiked with pyrene (the initial concentration ranged 1–75 μg/L), and shaken horizontally in the dark at 125 rpm for 31 days in order to achieve equilibrium (*Kah et al., 2011*). The POM strips were then removed from the vials and wiped with a wet tissue. The internal standard (pyrene-d10) was added and the POM strips were extracted with 20 mL of hexane by shaking for three days at 125 rpm (*Cornelissen & Gustafsson, 2004*). Extracts were concentrated under $N_2$ flow and analyzed by GC-MS. Sorption was measured by the POM method at 5–6 concentration levels and in duplicates.

## Data analysis

Sorption isotherms were fitted with the Freundlich model:

$$C_s = K_f \cdot C_w{}^n \tag{1}$$

where $C_s$ (μg/kg) is the sorbed concentration, $C_w$ (μg/L) is the concentration remaining in solution, $K_f$ ((μg/kg)/(μg/L)$^n$) is the Freundlich affinity and $n$ (-) is the linearity parameter. The model fitted all isotherms adequately, and there was no need to consider more complex models (potentially bringing a risk of over parametrisation, *Kah et al., 2011*). Sorption coefficients ($K_d$, L/kg) were calculated for $C_w$ 0.02 and 2 μg/L with the aim to compare the different sorbents at the low and high concentration levels included in the isotherms, respectively. Normalized coefficients, $K_{OC}$ values were also calculated based on the organic carbon content of each sorbent and sorbent mixtures (Table S3).

The dual-mode model was also fitted to the sorption isotherms in order to quantify the contributions of adsorption $Q_{ad}$ (μg/kg) and partitioning $Q_p$ (μg/kg):

$$C_s = Q_{ad} + Q_p = (Q_{max} * C_w)/(b + C_w) + K_p * C_w \tag{2}$$

where $Q_{max}$ (μg/kg) is the adsorption maximum capacity, b is the affinity coefficient (μg/L) and $K_p$ is the partitioning coefficient (L/kg).

All statistical analyzes, curve fits and graphs were produced using GraphPad Prism 6 (2016; GraphPad Software Inc., San Diego, CA, USA), with a significance level set to α = 0.05.

# RESULTS

## Sorption of pyrene to four biochars

Sorption isotherms of pyrene to the four biochars could be well fitted by the Freundlich model ($0.92 < r^2 < 0.97$, Fig. 1, all parameters are presented in Table S4). The linearity parameter n ranged between 0.45 and 0.67, which is characteristic of the highly nonlinear sorption of pyrene to carbonaceous sorbents (*Kah et al., 2016*). Nonlinearity increased with the pyrolysis temperature and was greater for the biochars derived from soft wood than from miscanthus ($p < 0.001$).

The biochars produced from miscanthus sorbed significantly more pyrene than biochars produced from soft wood, which was unexpected in view of the greater OC%,

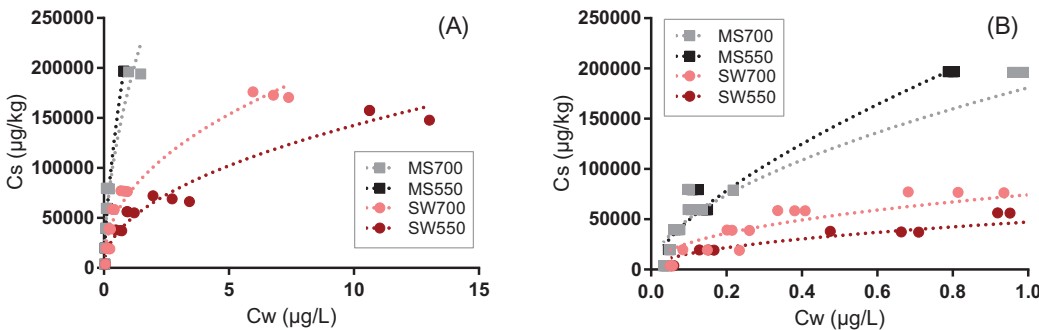

**Figure 1 Sorption of pyrene to four types of biochars produced from soft wood (circles) and miscanthus grass straw (squares) at 550 °C and 700 °C (SWP550, SWP700, MSP550 and MSP700, respectively).** The lines represent the fits with the Freundlich model (all fitted parameters are presented in Table S4). Graph (B) is a zoom of graph (A).

surface area and pore volume of the latter (Table S2). Sorption of organic sorbates generally increases with the biochar production temperature (*Kah et al., 2016*), and this was observed for soft wood, but not for miscanthus in our study. The discrepancy cannot be interpreted in terms of surface chemistry (e.g., aromaticity and polarity indices shown in Table S2) and neither in terms of physical parameters (surface area or porosity). *Behazin et al. (2015)* have previously shown that wood chip biochars exhibit atypical characteristics relative to miscanthus biochars, possibly related to differences in their structure, as shown on the electron microscope images (Figs. S1 and S2). Identifying the exact cause(s) for discrepancies in behavior between the two feedstock materials will require further investigations. Biochar MS550, which exhibited the highest sorption for pyrene, was considered for further investigations and it is designated as "biochar" in the following discussion.

## Comparison of the batch and POM methods

We had hypothesized that the compost would generate dissolved organic matter that can sorb significant amounts of pyrene. A previous study carried out with the same materials showed that concentrations in dissolved organic carbon increased by 40% for the clay loam and 100% for the sandy loam upon compost addition, reaching concentrations of about 850 mg/kg (details are available in *Bielská et al., 2017*). Dissolved organic matter may not be fully separated from the aqueous phase during the centrifugation step applied in the batch method, which can lead to $K_d$ batch < $K_d$ POM (the POM method only considers the freely dissolved portion of pyrene). Fig. S3 shows that the data generated by the batch and POM methods were consistent, and the isotherms generated in the low (with POM) and higher concentration range (by batch) constituted a continuous isotherm. When considering all sorbents, there was no significant difference in the $K_d$ values derived by the two methods (paired *t*-tests on $K_d$ values calculated based on isotherms fits at 0.05, 0.08, 0.2 and 0.5 µg/L, representing the range of concentration where both methods could be applied). When differences were noticed, $K_d$ values generated by batch tended to be greater than $K_d$ generated by POM, which goes against the hypothesis that dissolved organic matter played a significant role on the sorption of

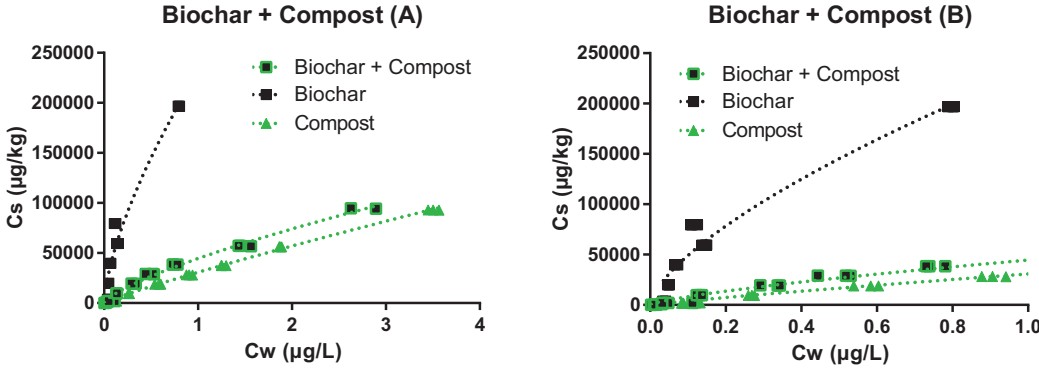

**Figure 2 Sorption isotherms of pyrene to the biochar (black squares), compost (green triangles), and their mixture (1:2, green and black squares).** Graph (B) is a zoom of graph (A).

pyrene. It was previously demonstrated that batch and POM methods can generate complementary results (*Kah et al., 2011*). In view of the general continuity of the isotherm generated by batch and POM (Fig. S3), the data were combined. The following discussion is thus based on isotherm analysis across several orders of magnitude.

## Sorption isotherms

All sorption isotherms were very well fitted with the Freundlich model ($r^2 \geq 0.95$, Figs. 2 and 3 all presented in the Table S5, and isotherms are presented in logarithmic scale in Figs. S4 and S5). The sorption affinity of pyrene to the clay loam was greater than to the sandy loam, which is consistent with the higher organic carbon content of the former.

The sorption isotherm to the pure biochar was the most nonlinear ($n = 0.67$), while sorption to the compost was one of the most linear ($n = 0.89$). The addition of compost to the biochar increased the $n$ value from 0.67 to 0.73 and brought the isotherm close to that of the compost (Fig. 2), indicating that significant fouling of the biochar sorption sites by the compost must have occurred (further discussed below). Similar observations were made with the soil mixture presented in Fig. 2: the linearity of the isotherms decreased upon the addition of biochar, and increased upon addition of compost (either with or without biochar). Changes in the isotherm linearity are consistent with previous literature, and reflect the contribution of partitioning to labile organic matter from soil and compost (close to linear), relative to the contribution of adsorption (nonlinear) to the carbonized surface of biochar (*Schwarzenbach, Gschwend & Imboden, 2017*; *Sigmund et al., 2017b*).

All sorption isotherms were also fitted with the dual mode model with the goal of distinguishing and quantifying the respective contributions of absorption (mainly driven by the soil and compost) and adsorption (mainly driven by the biochar). In many cases, the fits were ambiguous, meaning that several set of values for the three parameters could equally well fit the curve. This is likely because the dual mode model is too complex (over-parameterized) for the isotherms, leading to highly unreliable values that should not be interpreted. The following discussion is thus solely based on isotherm fits with

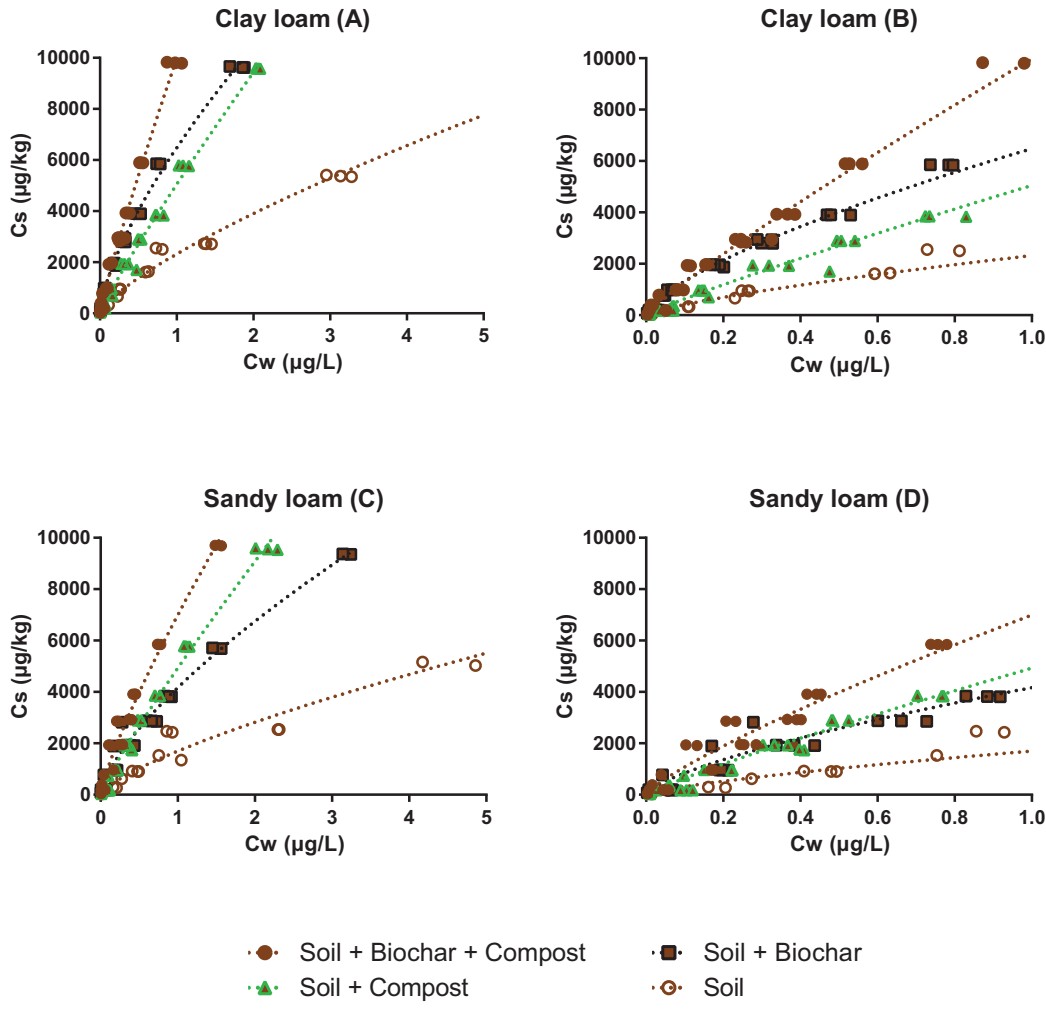

Figure 3 Sorption isotherms of pyrene to two soils before (open brown circles) and after addition of 5% biochar (black squares), 10% compost (green triangles) or both (full brown circles). Graph (B) and (D) are zooms of graphs (A) and (C), respectively. Note that the highest concentrations measured for the soils alone are not shown, but they were considered in the isotherm fit.

the Freundlich model. Differences in sorption affinity cannot be discussed based on $K_f$ values due to great differences in isotherm nonlinearity. Using the isotherm parameters presented in Table S5, sorption coefficients were calculated at two different concentrations and are discussed in the next section.

## Impact of amendment mixtures on sorption: $K_d$ values

Sorption coefficients calculated at $C_w = 0.02$ and 2 mg/L ($K_d0.02$ and $K_d2$) are presented in Fig. 4 (grey crosses, all values are available in Table S5) to allow comparisons of sorption among sorbents and their mixtures. Predictions of $K_d$ values for the mixtures were also computed based on the $K_d$ value of pyrene to each individual sorbent, and their respective proportion in a given mixture. The theoretical contributions of the soil, compost and biochar are represented by the brown, green and black bars in Fig. 4

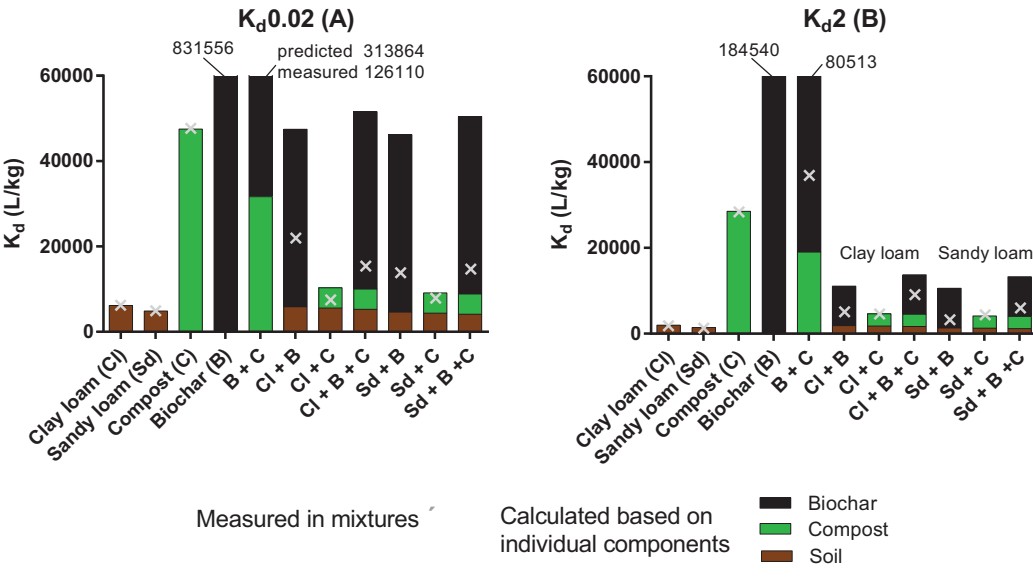

**Figure 4 Measured sorption coefficients (×) of pyrene at 0.02 mg/L (A) and 2 mg/L (B) compared to $K_d$ values calculated with an additive model (whole bars).** The bars show the contribution of soil (brown portion), compost (C, green portion) and biochar (B, black portion).

respectively. The top of the bar indicates the $K_d$ value predicted for the mixture by the additive model. For single sorbents, the prediction matches the measurement (100% contribution of one sorbent). For the mixtures, differences between the predicted (bars) and measured $K_d$ values (crosses) indicate the extent to which sorption was reduced through the interactions between sorbents.

### Compost + Biochar

Consistently with isotherm nonlinearity, $K_d0.02 >> K_d2$ for the biochar ($n = 0.67$), whereas sorption varied only marginally with concentration for the compost ($n = 0.89$). In the low concentration range, sorption to the mixture compost + biochar (2:1) was 2.7-fold greater than sorption to compost alone. Despite the great increase measured upon biochar addition, $K_d$ value in the mixture compost + biochar was 2.4-fold lower than that predicted based on the additive model. Assuming that sorption to biochar was hindered by compost (but not vice versa), the data suggests that sorption to biochar was reduced by 60% through interactions with the compost, much likely involving surface coverage and pore blocking (*Kwon & Pignatello, 2005*; *Pignatello, Kwon & Lu, 2006*; *Oen et al., 2012*). Trends were the same at the high concentration level, though slightly attenuated (e.g., reduction in sorption by biochar by 55%).

### Soil amendments with compost and biochar

The impact of pyrene concentration on sorption was the greatest in the soil + biochar mixtures ($K_d$ values were about four fold higher at 0.02 µg/L than at 2 µg/L), which is consistent with the strong isotherm nonlinearity. In the low concentration range, $K_d$ values increased in the order soil < soil + compost < soil + biochar + compost

< soil + biochar. The amendment of soil by biochar increased the sorption of pyrene by a minimum of 2.5 fold, reflecting the powerful enhancing effect that biochar can have on sorption, especially at low sorbate concentration. Soil amendments with compost only had moderate effects, increasing the sorption by 1.2 and 1.6 fold in the clay loam and in the sandy loam, respectively. As the concentration of pyrene increased, $K_d$ values followed the order soil < soil + compost $\leq$ soil + biochar < soil + compost + biochar, which is consistent with the OC% of the mixtures, and indicates that the specific sorption of pyrene to biochar surfaces was progressively tampered by the presence of soil and compost.

Predictions of sorption based on the additive model (bars on Fig. 4) overestimated the measurements (crosses) in almost all cases, but the degree of discrepancy depended very much on the type of sorbent mixture. Sorption of pyrene to the soil and to the compost is expected to be mainly driven by partitioning phenomenon (*Schwarzenbach, Gschwend & Imboden, 2017*) and estimates of sorption based on the additive model gave reasonable estimates of the sorption potential in the soil + compost mixtures (within a factor of 1.4 or less, see Fig. 4). Conversely, sorption affinity was generally greatly overestimated by the additive model in the mixtures containing biochar, especially at low concentration of pyrene. For instance, sorption to the soil + compost + biochar mixture was overestimated by about 3.5 times at 0.02 mg/L. The sorption of pyrene to pure biochar mainly consists in adsorption phenomena (typically >90%) and only very little contributions of partitioning (*Kah et al., 2016*). The discrepancies between the predictions and measurements suggest that mixing the adsorption (biochar) and partitioning-dominated phases (soil and compost) results in a transition from surface adsorption to partitioning processes as soil and/or compost are added (as indicated by changes in isotherm nonlinearity). Following the same reasoning as above, and assuming that sorption to biochar was hampered by the soil and compost (but not vice versa), the data obtained at low concentration indicates that sorption to biochar was attenuated by 81% and 84% in the presence of compost and clay loam or sandy loam, respectively.

## DISCUSSION

Many studies have reported on the extremely strong sorption potential of biochar towards contaminants based on experiments carried out with single sorbent at low concentrations. While such studies are valuable to elucidate the mechanisms of sorption as well as the relationships between biochar properties and functionalities, it is essential to note that sorption in the environment is likely to be strongly affected by a range of ageing processes. Our results are consistent with previous studies which have shown that biochar ageing in soils can greatly reduce its sorption potential (*Hale et al., 2011*; *Kumari et al., 2014*; *Wu et al., 2017*). The processes are likely to involve those elucidated by studies carried out on other carbonaceous materials and that showed that natural organic matter can bind to carbonaceous surfaces, block some of the pores and compete with organic contaminants for sorption sites (*Li et al., 2003*; *Chen, Chen & Zhu, 2008*; *Qiu et al., 2009*; *Zhang et al., 2012*; *Hou et al., 2013*; *Kah, Zhang & Hofmann, 2014*). More research will be needed to elucidate whether the attenuation of sorption is a truly competitive process,

or whether it is a kinetic process that reduces over time (as suggested for sediments, e.g., *Hale et al., 2009*).

In our study, the sorption coefficients derived from an additive model based on data obtained at high pyrene concentrations were within a factor two to the measurements in the soil + compost + biochar mixtures. For very pragmatic purposes (e.g., range finding), our results thus indicate that sorption of pyrene to mixtures may be predicted based on values measured for single sorbent at high sorbate concentration (here 2 mg/L). In the highest portion of the sorption isotherm, the high energy sorption sites of biochars are unlikely to play a significant role, which in some way accounts for the fouling of sorption sites by labile organic matter from e.g., soil and/or compost. This approach should be applied with great care as it does not specifically account for the processes discussed above, and it gives no indications about the bioavailability or desorption potential of the contaminant.

Another practical observation emerging from our study is that the soil amendments had very similar effects in the clay loam and the sandy loam, despite their different texture (15–27% clay) and OC content (1–1.7%). Previous studies have shown that soil physico–chemical properties and management practices (e.g., application rate and repetition) can affect the impact that biochar has on the sorption of contaminants to soil (*Kumari et al., 2014*). Our results however indicate that additions of similar quantities of compost and/or biochar could be advocated to achieve contaminant sequestration in soils having properties within the range investigated here.

## CONCLUSIONS

Detailed isotherm analyzes across a wide range of concentration confirmed that soil amendment with compost and biochar can significantly increase the sorption potential towards pyrene. The dissolved organic matter of the compost did not seem to play a significant role on the sorption of pyrene. Results were very similar in the two soils considered, and collectively suggest that combined amendments with compost and biochar may be a useful approach to remediate soils that are contaminated with relatively low levels of polycyclic aromatic hydrocarbons.

The addition of compost to soil led to a moderate increase in sorption (<2-fold), which could be well predicted based on measurements of sorption to the individual components. The results thus indicate that sorption of pyrene to soil and compost can be relatively well approximated as being additive phases dominated by partitioning phenomena. The addition of both compost and biochar to soil led to a major increase in sorption (2.5–4.7-fold), which was however much smaller than that suggested based on sorption to the three individual components. Results suggest that the strong sorption to the biochar alone was attenuated by up to 84% in the presence of soil and compost.

In realistic settings where biochar is amended to contaminated soil, biochar is subject to a range of biological, physical and chemical processes involving its interactions with various soil components, and leaching of its most soluble fractions (*Hale et al., 2011*; *Wang et al., 2017*). These processes can have a direct impact on the surface chemistry and porosity of biochar and the way it interacts with contaminants. The exact mechanisms

involved and the role they play in a given system remain mostly unknown. For instance, it has been suggested that ageing in soil has a limited impact on the pore size distribution of some biochars (*Sorrenti et al., 2016*; *Sigmund et al., 2017a*), but it is not known whether this applies to all biochar-soil systems. Our results show that soil and compost can significantly reduce the sorption of contaminants to biochar, and it has also been shown that pore blockage can entrap sorbed contaminants and reduce their desorption potential (*Yang & Xing, 2007*; *Wang et al., 2017*). How ageing and weathering processes affect the remobilisation of contaminants (desorption) has been relatively poorly considered up to now, although it an essential aspect impacting the long term performances of soil amendments. Lessons can be learnt from previous research carried out on the treatment of contaminated sediments with activated carbon (*Patmont et al., 2015*), but aspects specific to ageing processes occurring in soil certainly require further attention. More realistic studies carried out over longer periods of time are needed to evaluate the long term viability of soil remediation approaches based on biochar addition, with or without the addition of compost.

## ACKNOWLEDGEMENTS

We would like to thank Thorsten Hüffer for his help in designing the POM experiments, and Vesna Micic for taking the electron microscope images (both University of Vienna). We are also grateful to Gerhard Soja (Austrian Institute of Technology) for providing the soil samples.

### Funding

The authors received no funding for this work.

### Competing Interests

Melanie Kah is an Academic Editor for PeerJ.

### Author Contributions

- Melanie Kah conceived and designed the experiments, analyzed the data, prepared figures and/or tables, authored or reviewed drafts of the paper, approved the final draft.
- Gabriel Sigmund conceived and designed the experiments, performed the experiments, analyzed the data, approved the final draft, (revision; no important content changes).
- Pedro Luis Manga Chavez performed the experiments, approved the final draft.
- Lucie Bielská performed the experiments, approved the final draft.
- Thilo Hofmann conceived and designed the experiments, contributed reagents/materials/analysis tools, approved the final draft, (revision; no important content changes).

### Data Availability

The raw data are provided in a Supplemental File.

## Supplemental Information

Supplemental information for this article can be found online at http://dx.doi.org/
10.7717/peerj.4996#supplemental-information.

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
