# Peer review of "Sorption to soil, biochar and compost: is prediction to multicomponent mixtures possible based on single sorbent measurements?"

_PeerJ, doi:10.7717/peerj.4996_

## Round 0.1 · original submission · Minor Revisions

The two reviewers are positively impressed with this piece of work. Still, they addressed a couple of details that would be worth taking into account, prior acceptance.

Looking forward to read your revised version

Reviewer 1 ·

Basic reporting

The paper „Sorption to soil, biochar and compost …“ by Kah et al. describes an important aspect of sorption studies that definitely should be considered in many investigations dealing with the immobilization of pollutants to a sorbent.

Experimental design

The experiments described in this study have been well designed and have resulted in useful insights about sorption processes in a multicomponent system. This is a considerable progress compared to the all too common one-factor sorption studies.

Validity of the findings

Though I am generally quite happy with the paper, I suggest that the authors might consider a few minor issues:
• The readers of the article might be interested to get a little more information about the rationale of the selected experimental design. What was the reason for selecting 5% biochar and 10% compost as additive to soil? A conclusive comparison of the sorbents is not made easier if varying amounts of the materials have been used. Another option would have been to add e.g. equivalent amount of organic carbon with the sorbent.
• In the abstract it would be useful to indicate not only the changes in pyrene sorption by the added sorbents to soil as a multiple of the soil sorption but also to define which amount of added sorbent has caused these changes.
• The authors rely heavily on the Freundlich model of sorption isotherms. Although it may well be that this is the most appropriate model for the specific experiments, it would be useful to mention why not alternative models like Langmuir or Dubinin-Raduskevic have also been tested for comparing the performance of different models.
• Figures 2, 3, 4: it is not very fortunate to use only dark colors to show different treatments. Black can hardly be distinguished from dark-green and brown. More contrast is recommended.
• Tables S4, S5: it is not very fortunate to use in both tables the abbreviation Std. as in one case standard deviation, in the other case standard error is meant. The usual abbreviations are s.d. and s.e.

Additional comments

No additional comments.

Reviewer 2 ·

Basic reporting

see below

Experimental design

see below

Validity of the findings

see below

Additional comments

This is a neat piece of work on the sorption of pyrene to biochar, soil and compost, and combinations thereof. The study's originality is in the combination of compost and biochar. The rest of the science presented, that binding at carbonaceous sorbents is less strong in the presence of soil or sediment (the "attenuation" effect), has been known for over 10 years.
The work is technically sound, with both passive samplers and batch sorption experiments.
Some issues I have with the study include;
1. My most important question is: where are the DOC concentrations? There is a lot of argumentation around them, and that they could possibly change apparent Kd-values, but pyrene is not so very hydrophobic, so one needs high DOC concentrations to increase aqueous concentrations of pyrene. How high were the concentrations (if they were not measured, prepare dummy samples and do measure them), and would any effect on overall sorption be expected, calculated based on assumed K_DOC values for pyrene?
2. Plot all sorption isotherms as log-log plots, now it is hard to see the low-concentration points
3. Do not apply a Langmuir-like dual-mode sorption model for the sorbents, the requirements for Langmuir sorption are not fulfilled (monolayer coverage, homogeneous sorption sites). Just use a dual mode Freundlich isotherm,
CS= fsoilKDCW + fsorbKF,sorbCWn_sorbent
And derive n_sorbent from the sorption isotherms for the pure sorbents.
4. Give the MS a grammar check, there were some faulty sentences, e.g. line 73-74, line 90, line 328
5. It would be nice to find out whether the observed attenuation is a "real" phenomenon or only a kinetic thing – cite Hale et al 2009 (Hale, S. E., Tomaszewski, J. E., Luthy, R. G., & Werner, D. (2009). Sorption of dichlorodiphenyltrichloroethane (DDT) and its metabolites by activated carbon in clean water and sediment slurries. Water research, 43(17), 4336-4346) who observed that the fouling of AC by organic matter with regard to DDT adsorption was a kinetic phenomenon- after 2-3 years the sorption to AC in the presence of sediment was the same as sorption to clean AC. It is probably not possible to stretch the current measurements for such long times, but at least cite and discuss the possibility that this is a temporary phenomenon.
6. Line 60: of course strong sorption to a sorbent inhibits availability for degradation, but the fractions unavailable for degradation are also unavailable for uptake. More nuanced discussion needed.
7. Line 91: "contribution of DOC to sorption" – depends what you want to know. If you look at health effects, stick to the POM-based free concentrations. If you want to investigate transport, the total concentrations including DOC-bound ones are needed.
8. Line 307-308: apart from that this sentence is a bit meaningless, I also disagree: measuring freely dissolved concentrations with passive samplers is not SO difficult…..

---

## Round 0.2 · accepted · Accept

Thank you very much for taking into account the comments from the reviewers, and congratulations for this very nice piece of work.